# Biomarker-guided duration of Antibiotic Treatment in Children Hospitalised with confirmed or suspected bacterial infection (BATCH): protocol for a randomised controlled trial

Cherry-Ann Waldron ,[1] Emma Thomas-Jones,[1] Jolanta Bernatoniene,[2] Lucy Brookes-Howell,[1] Saul N Faust,[3,4] Debbie Harris,[1] Lucy Hinds,[5] Kerenza Hood,[1] Chao Huang,[6] Céu Mateus,[7] Philip Pallmann ,[1] Sanjay Patel,[3,4] Stéphane Paulus ,[8] Matthew Peak,[9] Colin Powell,[10,11,12] Jennifer Preston,[13] Enitan D Carrol[14]

For numbered affiliations see end of article.

**Correspondence to**
Dr Cherry-Ann Waldron;
waldronc@cardiff.ac.uk

## ABSTRACT

**Introduction** Procalcitonin (PCT) is a biomarker more specific for bacterial infection and responds quicker than other commonly used biomarkers such as C reactive protein, but is not routinely used in the National Health Service (NHS). Studies mainly in adults show that using PCT to guide clinicians may reduce antibiotic use, reduce hospital stay, with no associated adverse effects such as increased rates of hospital re-admission, incomplete treatment of infections, relapse or death. A review conducted for National Institute for Health and Care Excellence recommends further research on PCT testing to guide antibiotic use in children.

**Methods and analysis** Biomarker-guided duration of Antibiotic Treatment in Children Hospitalised with confirmed or suspected bacterial infection is a multicentre, prospective, two-arm, individually Randomised Controlled Trial (RCT) with a 28-day follow-up and internal pilot. The intervention is a PCT-guided algorithm used in conjunction with best practice. The control arm is best practice alone. We plan to recruit 1942 children, aged between 72 hours and up to 18 years old, who are admitted to the hospital and being treated with intravenous antibiotics for suspected or confirmed bacterial infection. Coprimary outcomes are duration of antibiotic use and a composite safety measure. Secondary outcomes include time to switch from broad to narrow spectrum antibiotics, time to discharge, adverse drug reactions, health utility and cost-effectiveness. We will also perform a qualitative process evaluation. Recruitment commenced in June 2018 and paused briefly between March and May 2020 due to the COVID-19 pandemic.

**Ethics and dissemination** The trial protocol was approved by the HRA and NHS REC (North West Liverpool East REC reference 18/NW/0100). We will publish the results in international peer-reviewed journals and present at scientific meetings.

**Trial registration number** ISRCTN11369832.

## Strengths and limitations of this study

► Trial will evaluate both safety and effectiveness of procalcitonin to guide antibiotic duration in children hospitalised with suspected or confirmed bacterial infection.
► Randomised controlled trial with multicentre design including patients from hospital sites in England and Wales. Efficient trial design with coprimary endpoints and including a health economic analysis and qualitative process evaluation.
► Due to the type of intervention it is not possible to blind patients and clinicians.
► Potential shift in patient population following a recruitment break due to COVID-19 pandemic will be addressed with sensitivity analyses.

## INTRODUCTION

Sepsis is defined as life-threatening organ dysfunction caused by a dysregulated host response to infection.[1] Sepsis causes many non-specific symptoms and signs that can also be caused by a large number of conditions that may or may not be due to infection, and that may or may not require immediate or urgent treatment. Sepsis is usually caused by bacteria, although viral and fungal causes do occur. The problem for clinicians is the difficulty in distinguishing bacterial sepsis from other conditions presenting with similar non-specific signs and symptoms. Prompt administration of antibiotics reduces mortality by half,[2] but indiscriminate antibiotic use unnecessarily increases antimicrobial resistance (AMR), resulting in increased

costs in hospitalised patients,[3 4] which represents a significant resource burden in the National Health Service (NHS). Not all children admitted with bacterial infection will meet the criteria for sepsis, but they could still have serious infection, requiring intravenous antibiotics for several days.

Biomarker blood tests currently used in the NHS, such as C reactive protein (CRP) do not reliably differentiate beween severe bacterial infection (SBI) (defined previously[5]) and inflammation, and show a delayed response to bacterial infection. Procalcitonin (PCT) is a biomarker released in response to inflammatory stimuli including bacterial infections, with very high levels produced in SBI.[6] In contrast to CRP, PCT rises early and peaks early, and falls rapidly in response to effective antimicrobial therapy. This makes blood PCT potentially a better biomarker for monitoring progression of SBI and response to antimicrobial therapy, and for facilitating clinical decision making by informing initiation, change or discontinuation of antimicrobial therapy. It supports the aims of the Department of Health Five Year Antimicrobial Resistance Strategy 2019–2024 of conserving and stewarding the effectiveness of existing antimicrobials by ensuring that antibiotics are used responsibly and less often (ie, antimicrobial stewardship (AMS)).[7]

There is strong evidence for introducing paediatric AMS programmes in hospital settings, in terms of reduced antibiotic use, improved quality of prescribing and cost-savings. Long-term and sustainable reductions in antimicrobial prescribing and a reduction of resistance rates at a population level have been achieved by the implementation of nationally coordinated, whole-system approaches, with no evidence of an increase in the rate of serious infection or bacterial complications.[8] We recently published a large study prospectively assessing the performance of multiple biomarkers of SBI in a heterogeneous cohort of critically ill children and uniquely profiled longitudinal biomarker changes. Longitudinal profiles for PCT showed the greatest percentage drop in values over the first 7 days of therapy in children with SBI, suggesting that PCT might be useful in guiding duration of antimicrobial therapy in children.[9]

National Institute for Health and Care Excellence (NICE) guidance on AMS (https://www.nice.org.uk/guidance/ng15) recommends decision support systems as an AMS intervention. The use of a PCT-based algorithm to guide antibiotic stopping or escalation is one such decision support system which can be used. The AMS guidelines made research recommendations including RCTs to determine whether short or long courses of antibiotics reduce AMR and whether using point-of-care tests is clinically and cost-effective when prescribing antimicrobials in children with respiratory tract infections. The Biomarker-guided duration of Antibiotic Treatment in Children Hospitalised with confirmed or suspected bacterial infection (BATCH) trial described in this paper is aligned with these recommendations in seeking to evaluate if PCT-guided management can result in shorter courses of antibiotics while not increasing the risk of adverse outcomes to paediatric patients.

A systematic review and cost-effectiveness analysis conducted on behalf of NICE (https://www.nice.org.uk/guidance/dg18) evaluated PCT testing to guide antibiotic therapy for the treatment of sepsis in intensive care settings and for suspected bacterial infection in emergency department (ED) settings in adults and children.[10] It concluded that addition of a PCT algorithm to the information used to guide antibiotic treatment may reduce antibiotic exposure in adults in intensive care unit (ICU) settings and in the ED without any adverse consequences and may also be associated with reductions in the length of hospital and ICU stay in adults. Evidence was not found on the effectiveness using a PCT algorithm to guide antibiotic treatment for children with suspected or confirmed SBI admitted from emergency care. None of the identified studies were conducted in the UK, and it was not clear whether the control arms of these studies were representative of standard practice in the UK. Therefore, the report recommended further studies to adequately assess the effectiveness of adding PCT algorithms to the information used to guide antibiotic treatment in adults and children with suspected or confirmed SBI in ICU settings and in adults and children with suspected bacterial infection in ED settings. It states that further studies are needed particularly for children, where data are currently lacking, and research examining (short-term) health-state utility values in the UK for adults and children with confirmed or suspected SBI in the ICU and ED.

A recent systematic review and meta-analysis of antibiotic duration for bacterial infections in children demonstrated that intravenous to oral switch can occur earlier than previously recommended.[11] The authors produced recommendations for antibiotic duration and intravenous to oral switch to support clinical decision making, and recommend prospective research on optimal antibiotic durations. The lack of good evidence on recommended duration of antibiotic therapy leads to an overuse of antibiotics, contributing to the development of AMR. Combating AMR has been identified as a national and global priority. Shorter courses of antibiotic therapy would be associated with reductions in adverse effects for patients, and reductions in healthcare resource utilisation. Results from this research will inform recommendations relating to the duration of antibiotic use in future guideline updates including NICE sepsis guidelines.

## Aims and objectives
The BATCH trial aims to improve AMS in hospitalised children with suspected or confirmed bacterial infection, by reducing antibiotic duration with guidance from an additional PCT laboratory test.

### Primary objective
To determine if the addition of PCT testing to current best practice based on NICE AMS guidelines can safely

allow a reduction in duration of intravenous antibiotic therapy in hospitalised children with suspected or confirmed bacterial infection compared with current best practice alone.

To meet this objective specifically, we will assess:

► Duration of intravenous antibiotics.
► Unscheduled admissions to paediatric intensive care unit (PICU) with infective diagnosis.
► Readmissions to PICU with infective diagnosis.
► Unscheduled readmissions with infective diagnosis within a week of stopping intravenous antibiotics.
► Re-commencing intravenous antibiotics for the same infection within a week of stopping intravenous antibiotics.
► Mortality.

### Secondary objectives

To assess the effect of additional PCT testing to AMS best practice on:

► Total duration of oral and intravenous antibiotics.
► Time to switch from broad spectrum to narrow spectrum antibiotics.
► Time to discharge from hospital.
► Hospital acquired infection (HAI).
► Suspected adverse drug reactions (ADR).
► Cost of hospital episode.
► Health-related quality of life.

Also to provide detailed understanding of parent and health professionals attitudes to and experiences of, participating in the BATCH RCT.

## METHODS AND ANALYSIS
### Design

This is a multi-centre, prospective, two-arm, individually randomised controlled trial. An internal pilot will assess the site and participant absolute recruitment and consent rates, the proportion of participants undergoing PCT assessments and the ability to collect the primary outcome data. Recruitment commenced in June 2018 and paused briefly between March and May 2020 due to the COVID-19 pandemic.

### Participants

Children admitted to hospital with suspected or confirmed SBI (diagnosed by the clinical team) and commenced on intravenous antibiotics, in whom intravenous antibiotics are likely to be continued for more than 48 hours. Recruited from paediatric wards or PICUs within children's hospitals and general hospitals in the UK (approximately n=10). The eligibility criteria are described in table 1.

### Intervention

In children randomised to the intervention arm, a blood sample will be sent to the hospital laboratory for a PCT test at baseline/randomisation and every 1–3 days while still on intravenous antibiotics to align with clinical workflow and routine laboratory testing where possible. This includes instances where intravenous antibiotics are restarted for the same infection (up to day 28 postrandomisation). An additional 1 mL (minimum 0.5 mL) lithium heparin samples will be collected for PCT analysis.

| Table 1 Eligibility criteria | |
|---|---|
| **Inclusion criteria** | **Exclusion criteria** |
| ► All children aged between 72 hours old and up to 18 years old admitted to hospital for confirmed or suspected severe bacterial infection (SBI), in whom intravenous antibiotics are commenced, and who are expected to remain on intravenous antibiotics for more than 48 hours.<br>► Conditions include (but not limited to): bacteraemia, central line-associated bloodstream infections (CLABSIs), uncomplicated bone and joint infections (such as single site infection, osteomyelitis with adjacent septic arthritis or septic arthritis with adjacent osteomyelitis), discitis, empyema, pneumonia, pyelonephritis, sinusitis, retropharyngeal abscess, pyomyositis, uncomplicated culture-negative meningitis, intra-abdominal infections, lymphadenitis, cellulitis.<br>► First time in the BATCH trial. | ► Preterm infants age <37 weeks corrected gestational age, under 72 hours old or ≥18 years of age.<br>► Children admitted moribund and not expected to survive more than 24 hours.<br>► Children with a predicted duration of intravenous antibiotics of less than 48 hours.<br>► Children not expected to survive at least 28 days because of a pre-existing condition.<br>► Children with bacterial meningitis,* bacterial endocarditis or brain abscess.<br>► Children with complicated bone and joint infections.†<br>► Children receiving antibiotics for surgical prophylaxis.<br>► Children with chronic comorbidities, such as cystic fibrosis, chronic lung disease, bronchiectasis where there is already a predefined length of course of antibiotics.<br>► Children who are severely immunocompromised (eg, chemotherapy, stem cell transplant, biological therapy for inflammatory or rheumatological conditions).<br>► Children who, in the opinion of the local investigator, are unsuitable for randomisation due to high probability of requiring sustained intravenous therapy.<br>► Children with a presence of existing directive to withhold life-sustaining treatment.<br>► Inborn infants admitted to neonatal intensive care units (NICU), neonatal high dependency units (NHDU), special care baby units (SCBU) or postnatal wards. |

*Excluded due to National Institute for Health and Care Excellence guideline on bacterial meningitis has predefined recommendations for duration of intravenous antibiotics.[22]

†Defined as chronic and/or related to a fracture or fixation device or prosthesis or implant. Chronic osteomyelitis presents six or more weeks after bone infection and is characterised by the presence of bone destruction and formation of sequestra.

BATCH, Biomarker-guided duration of Antibiotic Treatment in Children Hospitalised with confirmed or suspected bacterial infection.

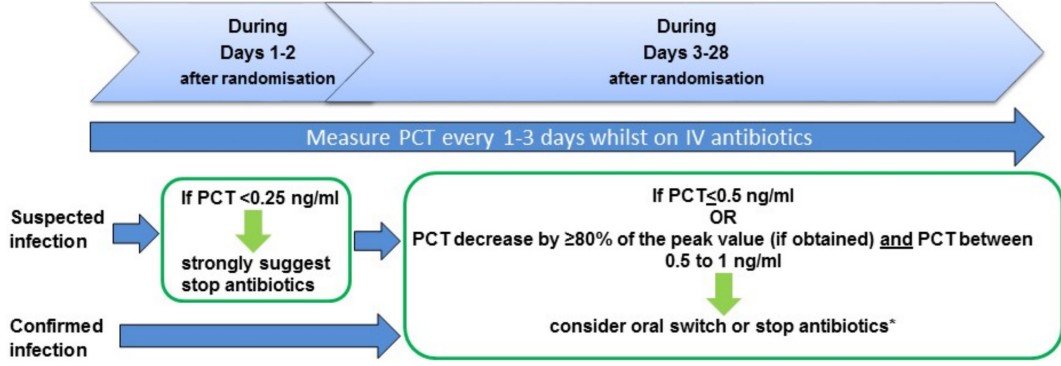

In the standard care group: use clinical response +/- CRP to guide oral switch and discontinuation.
In PCT group: use clinical response (+/- CRP) and PCT to guide oral switch and discontinuation.
**Measure PCT at randomisation/baseline and every 1-3 days whilst on IV antibiotics, or up to 28 days, as indicated clinically.** If on Outpatient Parenteral Antimicrobial Therapy (OPAT), frequency can be every 7 days or according to local standard care. PCT results will be made available to the clinician.

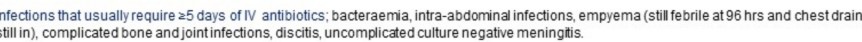
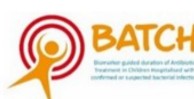

**Figure 1** Guidelines for continuing or stopping intravenous antibiotics. CRP, C reactive protein; PCT, procalcitonin.

PCT tests will be performed on a bioMérieux VIDAS platform. It is a prerequisite that participating sites have access this platform to take part in the trial. This is a semi-automated immunoassay system based on Enzyme Linked Fluorescent Assay principles. Calibration is performed in line with manufacturer's guidelines. It is simple and flexible to use and gives results in 20 min. It requires 200 µL of plasma or serum and can be run on a sample sent for routine biochemistry after the routine tests have been performed.

PCT results feed into an algorithm where thresholds have been defined by previous data (figure 1). The algorithm provides both definitive guidelines, for example, stop antibiotics if PCT<0.25 ng/mL, and advisory guidelines, for example, consider oral switch if PCT decreased by ≥80%. Clinicians can overrule the algorithm if they feel it is appropriate to do so. The algorithm values are based on our now published work which is the largest study to prospectively assess the performance of multiple biomarkers of SBI in a heterogeneous cohort of critically ill children and uniquely profiles longitudinal biomarker changes within the cohort.[9]

Children in the control arm will not have the PCT test performed.

## Primary outcome measure
The trial will use a coprimary outcome of antibiotic use and safety.
► Antibiotic usage is defined as the number of days intravenous antibiotics used.
► Safety is defined as the absence of all of the following:

– Unscheduled admissions/readmissions (to include readmission within 7 days of discharge with infective diagnosis, unscheduled readmission to PICU with infective diagnosis or admission to PICU with infective diagnosis).
– Retreatment for same condition within 7 days of stopping intravenous antibiotics (restarting intravenous antibiotics which have been stopped).
– Death.

## Secondary outcome measures
► Total duration of antibiotics (intravenous and oral).
► Time to switch from broad spectrum to narrow spectrum antibiotics.
► Time to discharge from hospital.
► Suspected ADR (categorised using the Liverpool Causality Assessment Tool comprising ten modified questions with their dichotomous responses into a flowchart to arrive at one of four outcome categories: 'definite', 'probable', 'possible' or 'unlikely'. It has high inter-rater agreement).[12 13]
► Cost of hospital episode.
► HAI as defined by the clinical team up to day 28.
► Health utility as measured by the Child Health Utility questionnaire (CHU9D)[14] up to day 28.

## Internal pilot
An internal pilot phase will be conducted over the first 8 months of the recruitment period with six lead sites. Predefined progression criteria will be used to assess feasibility to progress to the full trial, such as site and patient

absolute recruitment and consent rate, proportion of patients undergoing PCT assessments and the ability to collect primary outcome data.

## Trial procedures
### Data collection
All data collection will be by electronic data capture using a bespoke database developed by the Centre for Trials Research (CTR) and hosted by Cardiff University secure servers. It is encrypted and accessed by individual username and password. Paper copies of all case report forms will be available. Essential documents will be kept securely in a locked cupboard, and at the end of the trial, will be archived at an approved external storage facility for 10 years.

### Data management
Details of data management procedures (such as checking for missing, illegible or unusual values (range checks)) will be specified in the BATCH Data Management Plan. Details of Monitoring procedures will be specified in the BATCH Monitoring Plan.

### Identification and screening
Identification of potential participants will be by the clinical care team, or the clinical members of the research team involved in care of children on the ward, or the general paediatric or infectious diseases teams involved in care of children on the ward. This includes a member of the research team visiting the wards where children with SBI are admitted to assess eligibility and screening admissions lists.

### Informed consent
Informed consent will be obtained by those suitably trained and on the delegation log. Parent/carer(s) of children (or the child if over the age of 16 or Gillick competent) will be given a participant information sheet (online supplemental file 1) about the trial and will have time to consider before being asked to sign the consent form (online supplemental file 2). The consent form includes storing samples for future research. Age appropriate information sheets will also be provided for children who are old enough to use them, and those deemed to have capacity will be asked to sign an age appropriate assent form, additional to their parent/carer's consent.

Once consented, participants will be allocated a unique identification number (participant ID). Separate informed consent will be taken for participation in the qualitative data collection.

### Withdrawal
Participants may withdraw consent to participate in any aspect of the trial, at any time. Declining to participate or withdrawing from the trial will not affect the care of the child.

### Randomisation and enrolment
Children will only be randomised if the clinician expects intravenous antibiotics will be prescribed for longer than 48 hours. This will typically be between 20 and 48 hours after admission, to fit with clinical work flow of ward rounds and phlebotomy times for routine blood tests.

Participants will be allocated to the trial arms in a 1:1 ratio using minimisation[15] via a secure (24 hours) web-based randomisation programme controlled centrally by the CTR. The covariates whose imbalance is to be minimised are age group and site. A random element will be added to the minimisation algorithm (as described in Altman and Bland[15]) to reduce the risk of predictability and subversion. Full details are provided in the BATCH randomisation strategy. To further minimise the risk of subversion, the specific age groups and the random element will not be disclosed to those involved in recruiting participants until the end of the recruitment period.

All participants will be enrolled in the trial from the date of randomisation until day 28 follow-up. Participants will be assessed until they are discharged from clinical care. Figure 2 shows the trial schema.

### During hospitalisation
Outcome data described in table 2 will be recorded daily by the research nurse for all recruited participants (up to and including Day 28, or until discharge). Observation and medication charts and medical notes will be reviewed. Assessments include antibiotic use, routine test results, PCT measurements and clinician adherence to the algorithm.

### 28-day follow-up
At day 28 (+2 week time window) parents will be contacted by telephone or email to ask about their child's healthcare resource use (eg, hospital admissions, other prescribed medicines, over-the-counter medicines, general practice (GP) attendance), direct non-medical costs (ie, travel costs, child care costs, expenses incurred while in hospital, self-reported lost earnings) and the child's health-related quality of life (ie, CHU9D). If efforts to contact them by phone or email are unsuccessful, a questionnaire booklet will be posted out with a prepaid envelope for return.

## Public and patient involvement
Patient and public involvement (PPI) will be sought throughout the research process, from conceptualisation to dissemination. An example of this is the active involvement of the Liverpool GenerationR Young People's Advisory Group (YPAG) in contributing to the design of this research.[16] The group consists of 24 young people aged between 8 and 21 years old who have worked with several researchers exploring the topic of developing tests to rapidly detect or diagnose SBI in children. YPAG members are well aware of the problems associated with diagnosing and treating sepsis and have discussed at length the issues associated with AMR and the need to

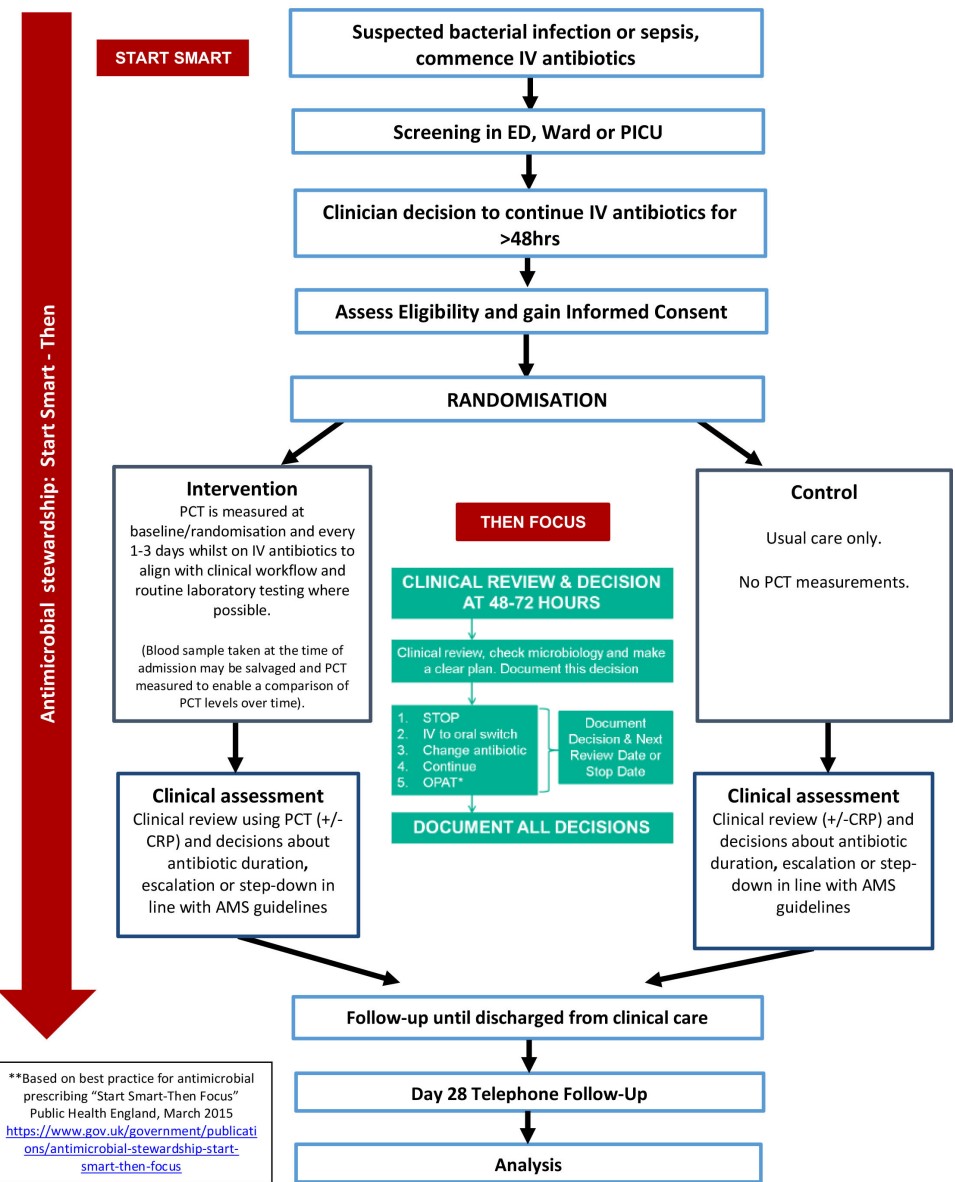

**Figure 2** Trial schema/participant flow. AMS, antimicrobial stewardship; CRP, C reactive protein; ED, emergency department; IV, intravenous; OPAT, Outpatient Parental Antibiotic Therapy; PCT, procalcitonin; PICU, paediatric intensive care unit.

educate young people and families about the misuse of antibiotics.

We will involve the Liverpool GenerationR YPAG, throughout the duration of the trial. The group will advise on children's information sheets and the production of educational materials for young people and families on the most appropriate use of antibiotics. We will invite parents and young people to contribute actively to dissemination events, including presenting parents' and young peoples' views and stories. Parent representatives will also be invited to attend Trial Management and Steering Group Meetings. Members of the YPAG and parents will be trained by our PPI lead.

### Safety reporting

Events described in table 3 (ie, hospitalisation and death) are primary outcomes of the trial and are recorded as part of primary outcome data collection, and therefore are not

subject to expedited reporting. Serious adverse events will be reported if the event results in persistent or significant disability or incapacity or consists of a congenital anomaly or birth defect. An assessment of causality between the event and the trial intervention will be carried out by the principal investigator or delegated clinician, and then independently by a clinical reviewer who will also assess expectedness. If the clinical reviewer classifies the event as probably or definitely caused by the intervention, it will be classified as a serious adverse reaction.

### Sample size

Two coprimary outcomes (intravenous antibiotics duration and a composite safety outcome) are defined in this trial[17] and the overall sample size is determined by both.

The focus for the intervention is on moving the step down from intravenous to oral therapy earlier, and therefore the time until this step down is the primary outcome

**Table 2** Schedule of enrolment, interventions and assessments

| | Data type | Source data | Data type | Screening | Baseline | Postrandomisation until discharged home | Follow-up (day 28) | Frequency | By whom |
|---|---|---|---|---|---|---|---|---|---|
| 1 | Informed consent | Consent form | – | X | | | | Once | Site clinical/ research team |
| 2 | Eligibility assessment | Eligibility Case Report Form (CRF) | – | X | | | | Once | Site research team |
| 3 | Demographics | CRF | | | X | | | Once | Research nurse |
| 4 | Admission data | CRF | Comorbidities, preadmission antibiotic use, initial working diagnosis | | X | | | Once | Research nurse |
| 5 | Health-related quality of life | Questionnaire | CHU9D | | X | | X | Twice | Patient/parent reported (over telephone or post at day 28) |
| 6 | Randomisation | CRF | – | | X | | | Once | Site research team |
| 7 | Antibiotic use (intravenous/oral) | Observation charts/medical notes | Antibiotic type, dose, duration | | | X | | Daily | Research nurse |
| 8 | Blood tests including PCT | CRF/medical notes | Routine blood tests PCT results (for those in intervention arm) | | | X | | As required | Research Nurse |
| 9 | Clinical review | CRF/medical notes | Clinical decision made and whether the algorithm was complied with | | | X | | As required when a clinical decision has been made | Site Clinical/ Research team |
| 10 | Cerebrospinal fluid metrics, radiology and microbiology | CRF/medical notes | White cell count, biochemistry. Microbiology results, radiology results | | | X | | As required | Research nurse |
| 11 | Re-commencing of Antibiotics (intravenous and oral | Observation charts/medical notes | Antibiotic type, dose, duration, time recommened | | | X | | Daily | Research nurse |
| 12 | Unscheduled admissions | Medical notes | PICU readmissions post discharge | | | X | | Daily | Research nurse |
| 13 | Mortality | Medical notes | Date, description | | | X | | If before day 28 | Research nurse |
| 14 | Discharge | Medical notes | Date, description | | | X | | If before day 28 | Research nurse |
| 15 | Adverse events | Observation charts/medical notes | Date, type | | | X | | Daily | Research nurse |
| 16 | Suspected adverse drug reactions (ADR) | Liverpool Causality Assessment tool | Date, description | | | X | | Daily | Research nurse |
| 17 | Resource use | Questionnaire | Direct medical costs (Inc. medication and ventilation and vasopressor) and resource use | | | | X | Once | Patient/parent reported (over telephone or post) |
| 18 | SAE | SAE form | | | ←-------------------------As required----------------------→ | | | | Research nurse |
| 19 | Withdrawals | Withdrawal form | | | ←-------------------------As required----------------------→ | | | | Research nurse, centre for trial research |

CHU9D, Child Health Utility questionnaire; PCT, procalcitonin; PICU, paediatric intensive care unit; SAE, serious adverse event.

**Table 3** Elements of the composite safety outcome

| Composite element | Definition | Reason for inclusion | Expected prevalence in usual care | Potential direction of change with intervention |
|---|---|---|---|---|
| Unscheduled admissions/ readmissions | Admitted/readmitted to PICU or unplanned readmission to hospital within 7 days of stopping intravenous antibiotics | Indicators of a deterioration and need for increased level of care | Our observation study showed 8.8% patients have admissions/readmissions[5] | Increase |
| Reinstating intravenous antibiotic therapy | Restarting intravenous antibiotic (for any reason) therapy within 7 days of stopping intravenous therapy | Indicator of potentially inappropriate withdrawal of intravenous antibiotics and deterioration | de Jong et al study 2.9% in control arm restarted intravenous antibiotic[23] | Increase |
| Mortality | Death for any reason in the 28 days following randomisation | – | PICANet Annual report 2015: deaths on PICUs ~4% in 2012–2014[18] | Increase |

PICU, paediatric intensive care unit.

on antibiotic usage, and the trial is powered to detect if PCT-directed care is *superior* to standard care on time until switch from intravenous antibiotics. The size of potential shortening of time to detect an effect has been taken from a systematic review.[10] The safety coprimary (table 3) is a composite measure reflecting various outcomes which represent deterioration or lack of clinical response in the child, and would be expected to increase if intravenous antibiotics were being withdrawn inappropriately early.

In terms of intravenous antibiotic duration, a 1 day reduction[18] in antibiotics from an estimated median of 5 days in the control arm (from our observation data[5]) demonstrates an HR of 1.25. At a 5% significance level with 90% power, 844 participants with observed intravenous antibiotics duration are needed. In terms of the event rates of safety elements, an observational study showed a readmission rate of 15% at day 28.[19] In critically ill patients, up to 3% reinstating intravenous antibiotic therapy rate, and 3% mortality were reported.[10 18 20] With some overlaps considered, we estimate around 15% overall rate of our composite safety outcome. The SAPS trial in adults used a non-inferiority margin of 8% for mortality.[10] Given the lower expected rate of safety outcomes in this population we have chosen a similar relative non-inferiority bound of 5%. This means increases in the composite safety measure of less than 5% (from 15% to 20%) using PCT guided therapy would be considered as not inferior. With a one-sided significance level of 0.05% and 90% power

we would need 1748 participants to test non-inferiority. Overall, with 1748 effectively recruited participants, we would have 99% power to detect antibiotic duration decrease and 90% power to test non-inferiority in safety separately. Assuming that these two coprimary outcomes are independent, this would give us at least 89% power for the combined analysis.[21] By considering 10% loss to follow-up for the primary outcomes, our final targeted sample size is inflated to 1942 as achievable.

## STATISTICAL ANALYSES
### Primary outcome
Our primary analysis of coprimary outcomes will be an intention to treat and will compare between trial arms: (a) the duration of days of intravenous antibiotics following randomisation using Cox regression and (b) the rate of adverse events using logistic regression, with a one-sided 95% CI constructed to assess non-inferiority. This analysis will control for balancing factors in the randomisation. A positive conclusion will only be made if both a decrease in intravenous antibiotic duration AND non-inferiority in safety can be demonstrated (table 4). We will assess if heterogeneity among centres exists and fit a two-level model with random centre effects if confirmed. Adherence to the PCT algorithm will be recorded, and per-protocol and complier average causal effect (CACE)

**Table 4** Combined primary outcome

| | Antibiotic duration different (reduction in PCT group) (H1) | Antibiotic duration no different (no reduction in PCT group) (H0) |
|---|---|---|
| Safety composite not worse in PCT group (H1) | ✔ | ✖ |
| Safety composite worse in PCT group (H0) | ✖ | ✖ |

✔—intervention successful; ✖—intervention unsuccessful.
PCT, procalcitonin.

**Table 5** Summary of analyses of coprimary outcomes

| | Coprimary outcomes | Analysis approach | Covariates in the model |
|---|---|---|---|
| Primary analysis | Duration of days of intravenous antibiotics (intervention effect) | Cox regression (superiority test) | Trial arm and minimisation factors (site, age group) |
| | Adverse events (composite safety outcome) | Logistic regression (non-inferiority test) | Trial arm and minimisation factors (site, age group) |
| Secondary analyses | Duration of days of intravenous antibiotics (intervention effect) | Kaplan-Meier plot | Trial arm |
| | | Log-rank test | Trial arm |
| | | Cox regression (assessments of suspected baseline confounders) | Covariates in the primary analysis, plus suspected baseline confounders (eg, gender) |
| | | Complier average causal effect (CACE) | Covariates in the primary analysis, plus intervention adherence |
| | Adverse events (composite safety outcome) | Logistic regression (assessments of suspected baseline confounders) | Covariates in the primary analysis, plus suspected baseline confounders (eg, gender) |

analysis will be undertaken to test the treatment effects to patients with fully used PCT algorithm (table 5). No interim analysis will be performed. The trial statistician will be blinded to the study group allocation.

### Secondary outcomes

For secondary outcomes, differences in the proportion of ADR, HAI, unscheduled readmission, re-commencing intravenous antibiotics and 28-day mortality will be assessed separately by logistic regression models. We will report the median (IQR) of the total duration of antibiotics (intravenous and oral), time to switch from broad to narrow spectrum antibiotics and time to discharge from hospital in both treatment groups and assess the group differences via Kaplan-Meier plots and Cox regression. Average utility will be compared between the two groups at 28 days using linear regression (table 6). The ineligible/inevaluable participants will be excluded in secondary analyses (per protocol, CACE, etc).

### Sub group analysis

Several exploratory subgroup analyses, including one split by the organ system of the infection (ie, lower urinary tract, lower respiratory, intra-abdominal, bacteraemia, skin and soft tissue, etc), will be prespecified in the Statistical Analysis Plan.

### Missing, unused and spurious data

Missing primary outcome data are likely to be minimal, so complete case analysis will be used. However, if this exceeds more than 20% of participants we will employ multiple imputation and report the impact on the treatment effect alongside the complete case analysis.

### Economic evaluation

Health economic analysis will include direct and indirect costs associated with unscheduled admissions (to ward or PICU), re-admissions, re-starting intravenous antibiotics and hospital-acquired infections. Descriptive and regression analysis will be used to identify key elements of service use and cost and to explore the potential impact of baseline participant characteristics on the costs and outcomes measures. Average cost per participant will be estimated at the end of the treatment and the follow-up periods, respectively, and average cost per subgroup of patients may be explored for the same time points. Bootstrapping

**Table 6** Summary of analyses of secondary outcomes

| Secondary outcomes | Analysis approach | Covariates in the model |
|---|---|---|
| Proportion of ADR<br>Proportion of HAI<br>Proportion of unscheduled readmission<br>Proportion of re-commencing intravenous antibiotics<br>Proportion of mortality | Logistic regression | Trial arm and minimisation factors (site, age group) |
| Duration of antibiotics (intravenous and oral)<br>Time to switch from broad to narrow spectrum antibiotics<br>Time to discharge from hospital | Cox regression | Trial arm and minimisation factors (site, age group) |
| Average utility | Linear regression | Trial arm and minimisation factors (site, age group) |

ADR, adverse drug reactions; HAI, hospital acquired infection.

and missing data imputation will be done if justified. Differences in each arm will be assessed and used for the computation of an incremental cost-effectiveness ratio (ICER). We will calculate ICERs for a clinically effective outcome (fewer days on intravenous antibiotics with increased or equivalent safety) and the cost per intravenous antibiotic day avoided.

A cost-effectiveness analysis will assess possible efficiency gains. An NHS perspective will be used and relevant direct medical costs will be collected. Information on resource use will include data on inpatient bed days, antibiotic consumption, nursing and medical resources, other medicines including over the counter medicines, diagnostic and monitoring laboratory tests, GP visits and emergency visits. Direct hospital costs will be calculated by multiplying resource use with the accompanying unit costs collected from patient level data in the participating hospitals, routine NHS sources (eg, NHS reference costs and *British National Formulary* and from the manufacturer of the PCT test, as appropriate. Time horizon will be 28 days, therefore there is no need to consider a discount rate. Patients' health utility will be measured using CHU9D up to day 28.

Descriptive and regression analysis will be used to identify key elements of service use and cost and to explore the potential impact of baseline participant characteristics on the costs and outcomes measures. Differences in each arm will be assessed and used for the computation of an ICER. One-way sensitivity analysis will be carried out in key model parameters. Probabilistic sensitivity analysis and cost-effectiveness acceptability curves will be constructed. Information on direct non-medical costs, such as travelling to and from the hospitals, and indirect costs, such as parents' productivity losses, will also be collected.

### Qualitative study

The qualitative evaluation will aim to explore the experiences and understanding of parents of children recruited into the BATCH trial (n=10–15) about their child's condition and treatment of confirmed or suspected bacterial infection, and also to explore their views and experiences about participating in an RCT. Interviews will be conducted after the participant reaches their day 28 follow-up. A purposive sample strategy will be employed to include parents from both intervention and control arms and inclusion of different sites.

It will also aim to explore the views and experiences of health professionals involved in the BATCH trial (n=10–20) about participating in an RCT with a focus on acceptability of the trial, clinical equipoise, taking informed consent and support needs of trial involvement. Interviews will be conducted at two time points (before or at the beginning and after the intervention) which will enable us to capture whether there are any changes in attitudes towards the PCT test. A purposive sample strategy will be developed to address representation from up to five different sites and variation in health professional role (eg, ward nurse, consultant, research nurse, etc).

With regards to the sample size for both health professionals and parents, the qualitative researcher(s) will make pragmatic decisions along with the research team regarding when enough is known about certain themes (ie, data saturation has occurred.

Semi-structured interviews will be undertaken face-to-face or via telephone. They will be recorded and transcribed. Data will be double coded and analysed using thematic analysis. Non-participant observation of episodes of patient care and trial delivery (of both arms) will also be carried out in some centres. Observations and field notes will enable an understanding of how the individual intervention components and delivery processes work in the real healthcare setting.

## TRIAL MANAGEMENT

The trial is sponsored by University of Liverpool, coordinated by CTR, Cardiff University and hosted by Alder Hey Children's NHS Foundation Trust. The other partner organisations are University of Southampton, University Hospital Southampton NHS Foundation Trust, University Hospitals Bristol NHS Foundation Trust, Oxford University Hospitals NHS Foundation Trust, Sheffield Children's NHS Foundation Trust, Lancaster University and Hull University.

### Trial Management Group

The Trial Management Group (TMG) will meet monthly throughout the course of the trial and will include the chief investigator coapplicants, collaborators, trial manager, data manager, statistician and administrator. A parent representative will also attend and contribute to the design and management of the trial, as well as patient-facing materials. TMG members will be required to sign up to the remit and conditions as set out in the TMG charter.

### Trial Steering Committee

A Trial Steering Committee (TSC) consisting of an independent chairperson, two independent members and a parent representative will provide oversight of the BATCH trial. Members will be required to sign up to the remit and conditions as set out in the TSC charters and will meet at least annually.

### Independent Data Monitoring Committee

An Independent Data Monitoring Committee (IDMC) will provide oversight of all matters related to patient safety and data quality. Members will be required to sign up to the remit and conditions as set out in the IDMC charter and will meet at least annually.

## ETHICS AND DISSEMINATION
### Research approvals

The trial was approved by the Health Research Authority and NHS Research Ethics Committee (North West Liverpool East REC reference 18/NW/0100) on 13 April 2018.

The following substantial amendments were made to the trial and were communicated to all trial sites: Amendment 2 (30 May 2018); Amendment 3 (13 July 2018); Amendment 5 (12 December 2018); Amendment 9 (9 August 2019); Amendment 14 (9 October 2020). The current protocol is version 5.0 dated 8 July 2020.

## Dissemination plan

Following completion of the trial, a final report will be prepared for the National Institute of Health Research (NIHR) HTA Journal series. Findings will be published in peer-reviewed journals and presented at international conferences. Nationally, we will engage with NICE, the Royal College of Paediatrics and Child Health, The British Society for Antimicrobial Chemotherapy, British Infection Society, and the British Paediatric Allergy, Immunity and Infection Group. All publications and presentations related to the trial will be authorised by the TMG in accordance with the BATCH publication policy. Where appropriate, the results of this trial can be directly implemented in the revisions of NICE guidelines.

## DISCUSSION

This trial has the potential to impact the clinical care of hospitalised children with confirmed or suspected SBI, which currently accounts for a large proportion of antibiotic use in hospitalised children. It will contribute to the development of PCT-guided antibiotic management guidelines of infections in hospitalised children, and will address the safety of shorter antibiotic courses. If shorter duration of PCT-guided antimicrobial therapy is shown to be safe and effective, this will have major implications for direct and indirect costs of childhood hospitalisations from infection. This will potentially lead to significant reductions in duration of hospitalisation and reduced antibiotic exposure, resulting in a positive impact on healthcare services and societal costs. Reduced exposure to antibiotics will, in turn, reduce AMR. This trial will demonstrate if introducing PCT-guided AMS management in the NHS is cost-effective. Reduced antibiotic use will lead to reduced hospital costs and reduced ADRs.

This trial is timely as it aligns with the current Department of Health Five Year action plan for AMR 2019 to 2024 and is a response to research recommendations from two published NICE guidance documents (DG18 and NG15).

### Author affiliations

[1]Centre for Trials Research, College of Biomedical and Life Sciences, Cardiff University, Cardiff, UK
[2]Department of Paediatric Infectious Disease and Immunology, Bristol Royal Hospital for Children, Bristol, UK
[3]NIHR Southampton Clinical Research Facility and Biomedical Research Centre, University Hospital Southampton, University of Southampton NHS Foundation Trust, Southampton, UK
[4]Faculty of Medicine and Institute for Life Sciences, University of Southampton, Southampton, UK
[5]Department of Paediatrics, Sheffield Children's NHS Foundation Trust, Sheffield, UK
[6]Hull York Medical School, University of Hull, Hull, UK
[7]Division of Health Research, Lancaster University, Lancaster, UK
[8]Department of Paediatrics, University of Oxford, Oxford, UK
[9]Alder Hey Children's NHS Foundation Trust, Liverpool, UK
[10]Department of Paediatrics, Children's Hospital for Wales, Cardiff, UK
[11]Sidra Medicine, Doha, Qatar
[12]Division of Population Medicine, School of Medicine, Cardiff University, Cardiff, UK
[13]NIHR Alder Hey Clinical Research Facility, Alder Hey Children's Hospital NHS Foundation Trust, Liverpool, UK
[14]Institute of Infection, Veterinary and Ecological Sciences, University of Liverpool, Liverpool, UK

**Acknowledgements** In addition to the authors, the BATCH team comprises Helen Nabwera, Katrina Cathie, Daniel Owens, Jenny Whitbread, Zoe Gray, Zoe Oliver, Judith Evans, Sarah Jones, Sam Clarkstone, Simon Schoenbuchner, Kim Smallman and Sarah Milosevic. The authors would like to thank the research nurses, clinicians and laboratory staff for their support during the trial. They would also like to acknowledge the contribution of the Liverpool GenerationR YPAG, the TSC members, namely Ann Marie Swart, Paul Brocklehurst, Colette Smith, Paul Dark, Phil Shackley, and Kelly Chapman, and IDMC members, namely Graeme MacLennan, Danielle Horton Taylor, Philip Howard and Alastair Sutcliffe. The CTR receives funding from Health and Care Research Wales and Cancer Research UK.

**Contributors** EDC is the chief investigator of this trial. EDC along with JB, SNF, SaP, LH, StP, ET-J, LB-H, CM, CH, CP, JP, MP and KH led the development of the research question, study design, obtaining the funding and implementation of the protocol. C-AW is the trial manager and ET-J is the senior trial manager who coordinate the operational delivery of the trial protocol and recruitment. LB-H is the lead qualitative researcher. CH and PP are the trial statisticians. DH is the data manager. All authors listed provided critical review and final approval of the manuscript.

**Funding** This trial was funded by the National Institute of Health Research Health Technology Assessment (NIHR HTA) Programme, reference 15/188/42. Sponsored by University of Liverpool, Research Support Office, 2nd Floor Block D, Waterhouse Building, 3 Brownlow Street, Liverpool, L69 3GL. Contact person: Mr Neil French; Sponsor@Liverpool.ac.uk, Sponsor reference: UoL001333. The study is supported by the NIHR Clinical Research Network, NIHR Southampton Clinical Research Facility.

**Disclaimer** The views expressed are those of the authors and not necessarily those of the NHS, the NIHR or the Department of Health and Social Care. Neither the Sponsor nor the Funder had any role on the study design; collection, management, analysis and interpretation of data; writing of this manuscript or in the decision to submit this manuscript for publication.

**Competing interests** EDC is a member of NIHR Invention for Innovation panel, member of NICE Diagnostic advisory committee April 2014–September 2020, Member of MRC DPFS panel and MRC COVID-19 Agile panel. SNF is an NIHR Senior Investigator. All other authors declare no competing interests.

**Patient consent for publication** Not applicable.

**Provenance and peer review** Not commissioned; externally peer reviewed.

**ORCID iDs**
Cherry-Ann Waldron http://orcid.org/0000-0001-8465-2492
Philip Pallmann http://orcid.org/0000-0001-8274-9696
Stéphane Paulus http://orcid.org/0000-0002-0703-9114

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
