## [Reviewer comments · BMJ Open]

ARTICLE DETAILS

TITLE (PROVISIONAL)	Biomarker-guided duration of Antibiotic Treatment in Children Hospitalised with confirmed or suspected bacterial infection (BATCH): Protocol for a randomised controlled trial.
AUTHORS	Waldron, Cherry-Ann; Thomas-Jones, Emma; Bernatoniene, Jolanta; Brookes-Howell, Lucy; Faust, Saul; Harris, Debbie; Hinds, Lucy; Hood, Kerenza; Huang, Chao; Mateus, Ceu; Pallmann, Philip; Patel, Sanjay; Paulus, Stephane; Peak, Matthew; Powell, Colin; Preston, Jenny; Carrol, Enitan

VERSION 1 – REVIEW

REVIEWER	Shehabi, Yahya Monash University
REVIEW RETURNED	21-Dec-2020

GENERAL COMMENTS	Thank you for asking to review this protocol manuscript. Well-written for a clear and a practical protocol. Minor comments: In the intro and study rationale, I suggest some justification of the cut-off values used. These values, mainly used in adults, may not be reflective of PCT kinetics in septic children. Perhaps collecting data such as paediatric organ dysfunction score and comparing that to baseline PCT may be a nice way to stratify patients. It will also allow subgroup analysis based on baseline PCT level. There is a suggestion in the literature that a PCT as high as 4 ng/ml is where the optimal cut-off for separating high severity septic kids from those of low severity score. PCT testing times, it would better if it is standardized, like day 1, 3 and 5 and then whenever clinically indicated till day 28. I suggest considering storing samples in the control group for batch PCT testing for comparison with intervention group. Process related outcomes and protocol compliance would be important outcomes. Are there plans to monitor compliance through the trial? This knowledge may improve the internal validity of the trial. Very minor point; In the secondary objectives, page 5, third point, please separate hospital acquired infection from time to discharge from hospital.
---

REVIEWER	Fontela, Patricia McGill University, Pediatrics
REVIEW RETURNED	05-Feb-2021

GENERAL COMMENTS	Summary:
----------

Thank you for the opportunity to review this protocol. I read it with great interest.

This is the protocol for a multi-centre, parallel, two-arm, randomized controlled trial that primarily aims to assess whether the addition of a procalcitonin-guided algorithm to current best practice can safely reduce intravenous (IV) antibiotic use duration (primary outcome) in hospitalized pediatric patients treated with intravenous antibiotics for suspected or confirmed bacterial infection, compared to the use of best practice only. Antimicrobial best practices are based on the National Institute for Health and Care Excellence Antimicrobial Stewardship (NICE AMS) guidelines. Safety outcomes will include: unscheduled PICU admissions with infectious diagnosis, PICU readmissions with infectious diagnosis, unscheduled hospital readmissions with infectious diagnosis within a week of stopping IV antibiotics, re-starting IV antibiotics for the same infection within a week of stopping IV antibiotics, mortality

Secondary study aims include to assess the effect of the proposed intervention on the total duration of antimicrobial treatment (IV and oral), time to switch from broad to narrow spectrum antibiotics, time to hospital discharge, incidence of hospital-acquired infections, incidence of suspected adverse drug reactions, cost effectiveness (cost of hospitalization episode), and health-related quality of life.

Authors also plan to perform an internal pilot study to assess site and participant recruitment and consent rate, as well as compliance with protocol (procalcitonin assessments) and the ability to gather primary outcome data. Finally, authors will also perform a qualitative evaluation of the process of participating in the study.

My major questions/comments relate to clarification of the timing of randomization, patient eligibility criteria, and the intervention. Other minor comments are listed below.

Major comments:

- Participants:

- o Is the diagnosis of the infectious processes based on the treating team opinion or will they be adjudicated by the research team? If adjudication will be used, more information on how it will be performed is needed.

- o Please provide definitions for “suspected” and “confirmed” SBI.

- o The rationale for excluding patients with bacterial meningitis is not presented. However, patients with uncomplicated culture-negative meningitis, which may be caused by viruses, will be included. Bacterial meningitis is an important severe infection in the pediatric patient population for which guidance regarding duration of IV antibiotic/total duration of antibiotic treatment would be very useful.

- o The definition of complicated bone and joint infections should be provided

- Identification and screening (Page 8, line 30):

- o More detail in this section must be provided. Please describe how potentially eligible participants at different study sites will be identified.

	 • Intervention (Page 6, line 41):  o What is the intended timing of randomization? Will patients be eligible to be randomized at any time of the antibiotic treatment (e.g., day 5) or during a specific time window? o Randomization: more details are needed. Based on the protocol, it seems that central randomization will be stratified by age group and site. What are the age groups that will be used? A random element will be added to the randomization algorithm: please clarify how this will be done. o Blinding: based on the protocol, I presume that patients/families and treating team will not be blinded to the intervention. However, will members of the research team be blinded for study group allocation (outcome assessors, statistician, etc)? o Procalcitonin measurement: authors state that procalcitonin will be measured at randomization and every 1-3 days. Will the frequency of procalcitonin test be defined by the treating team or will it be based on the study protocol? More clarification is needed. o In addition, procalcitonin will be measured using the bioMérieux VIDAS platform. Do all sites have access to it? What is the test turnaround time? Will the same machine models and calibration parameters be used at all sites? What is the measuring range of the platform that will be used? Will you use serum or plasma? o Procalcitonin algorithm: the rationale behind the procalcitonin algorithm that will be used in the study should be discussed in detail. It is important to specify that the thresholds that will be used were obtained from previous adult studies and not from pediatric patients. • Outcomes:  o Mortality: please define the type of mortality that will be studied (e.g., hospital mortality, 30-day mortality, etc). o Hospital-acquired infections (HAI): how will the diagnosis of such infections be made? Will they be based on the treating team diagnosis? Or will they be adjudicated by the local infection control team? Are the HAI definitions used in different participating sites the same (or at least similar)? o Suspected adverse drug reactions: it is very difficult to establish a causal relationship for an adverse reaction, especially when patients are taking multiple medications. How will this outcome be adjudicated? • Sample size (Page 11, line 11):  o The authors have referred to the primary outcome outcome throughout the manuscript as “duration of IV antibiotics,” but could consider renaming it, e.g., “time in days until IV antibiotic cessation”, to more clearly align with the statistical analysis (Cox model) chosen. o Non-inferiority margin: non-inferiority margin for a composite safety outcome that includes mortality was set at 5%. Earlier procalcitonin RCTs were criticized because the non-inferiority margin for mortality was considered too high. More clarification regarding the rationale behind the 5% non-inferiority margin is needed. • In line with the previous comment, have the authors considered also reporting measures on the time scale (e.g., comparing median durations)? This can be useful in cases when the event of interest (switching IV to oral) will eventually occur in all patients.
--	---

	 • Statistical analysis:  o As the hazard ratio depends on length of follow-up, authors should clarify what follow-up duration (e.g., 28 days) is being considered for the Cox regression for the co-primary outcome (time to switch from IV to oral). o Secondary outcomes: The secondary outcomes listed on Page 7 include 1) time to switch from broad to narrow spectrum antibiotics, and 2) hospital acquired infection up to day 28. However, the statistical analysis for these outcomes is missing. o Interim analysis: it is not clear if an interim analysis will be performed. Information about when it will be performed, as well as the statistical methods that will be used for it, must be provided. • Qualitative study – methods: what is the theoretical framework that will be used to analyze the qualitative data? How long after the intervention will interviews be conducted? How will participants be selected? How will final sample size be defined? • Internal pilot: It is unusual to perform a pilot study embedded in the final RCT. It is commonly performed before the RCT to generate data on study feasibility. Questions arise regarding the internal pilot study: when are the planned dates for the internal pilot? How many and which sites are being included in the pilot trial? What happens if data show low proportion of internal pilot outcomes (e.g., low proportion of patients undergoing procalcitonin measurement, low proportion of patients for whom primary outcome data were collected, etc)? Will the RCT be considered as non-feasible? More clarification about how the results of the internal pilot study will be used in needed. Minor comments:  • Methods - Setting: how many hospitals will be participating in this RCT? It is not clear in the protocol. • Page 3, line 17: CRP not previously defined. • Page 6, line 52: Authors should state for ethical reasons the amount of blood sample needed from each child to achieve the 200µl of plasma. • Secondary outcomes (Page 7, line 43): Please provide a reference for both the Liverpool causality assessment tool and the CHU9D instrument, and a description of their reliability/validity. • Page 8, line 14: Please describe how the paper copies will be kept confidential and for how long. • Page 9, line 6: Will authors seek to replace subjects who withdraw in order to maintain adequate sample size? • Page 9, line 17: Please define age groups. • Page 12, line 21: Small typo. Please remove “. are observed”. • Page 23, line 27: Missing a word after “Antibiotic type, dose, duration, including...”
--	---

VERSION 1 – AUTHOR RESPONSE

Reviewer: 1

Dr. Yahya Shehabi, Monash University

Comments to the Author:

Thank you for asking to review this protocol manuscript.

Well-written for a clear and a practical protocol.

Comment	Response
In the intro and study rationale, I suggest some justification of the cut-off values used. These values, mainly used in adults, may not be reflective of PCT kinetics in septic children. Perhaps collecting data such as paediatric organ dysfunction score and comparing that to baseline PCT may be a nice way to stratify patients. It will also allow subgroup analysis based on baseline PCT level. There is a suggestion in the literature that a PCT as high as 4 ng/ml is where the optimal cut-off for separating high severity septic kids from those of low severity score.	This has been added to the Intervention section (p.7). These values are based on our now published work which is the largest study to prospectively assess the performance of multiple biomarkers of SBI in a heterogeneous cohort of critically ill children and uniquely profiles longitudinal biomarker changes within the cohort. Nielsen MJ, et al. Procalcitonin, C-reactive protein, neutrophil gelatinase-associated lipocalin, resistin and the APTT waveform for the early diagnosis of serious bacterial infection and prediction of outcome in critically ill children. PLoS One. 2021 Feb 5;16(2):e0246027. doi: 10.1371/journal.pone.0246027. PMID: 33544738; PMCID: PMC7864456.
PCT testing times, it would better if it is standardized, like day 1, 3 and 5 and then whenever clinically indicated till day 28. I suggest considering storing samples in the control group for batch PCT testing for comparison with intervention group.	This has been clarified in the methods (p.6) The PCT testing times are not standardised to allow for testing to align with clinical workflow and routine laboratory testing where possible. We are storing samples in the control group as part of the PRECISE sub-study NIHR Funding and Awards Search Website
Process related outcomes and protocol compliance would be important outcomes. Are there plans to monitor compliance through the trial? This knowledge may improve the internal validity of the trial.	Clinicians' adherence to the PCT algorithm will be recorded (see subsection 'During hospitalisation' (p.10) and Table 3). Per-protocol and complier average causal effect (CACE) analyses will be performed to assess the effect of the intervention in the subgroup of participants for whom the PCT algorithm was utilised as intended (see subsection 'Primary outcome' (p.13) and Table 5).
In the secondary objectives, page 5, third point, please separate hospital acquired infection from time to discharge from hospital.	This has been done.

Reviewer: 2

Dr. Patricia Fontela, McGill University

Comments to the Author:

Summary:

Thank you for the opportunity to review this protocol. I read it with great interest.

This is the protocol for a multi-centre, parallel, two-arm, randomized controlled trial that primarily aims to assess whether the addition of a procalcitonin-guided algorithm to current best practice can safely reduce intravenous (IV) antibiotic use duration (primary outcome) in hospitalized pediatric patients treated with intravenous antibiotics for suspected or confirmed bacterial infection, compared to the use of best practice only. Antimicrobial best practices are based on the National Institute for Health and Care Excellence Antimicrobial Stewardship (NICE AMS) guidelines. Safety outcomes will include: unscheduled PICU admissions with infectious diagnosis, PICU readmissions with infectious diagnosis, unscheduled hospital readmissions with infectious diagnosis within a week of stopping IV antibiotics, re-starting IV antibiotics for the same infection within a week of stopping IV antibiotics, mortality

Secondary study aims include to assess the effect of the proposed intervention on the total duration of antimicrobial treatment (IV and oral), time to switch from broad to narrow spectrum antibiotics, time to hospital discharge, incidence of hospital-acquired infections, incidence of suspected adverse drug reactions, cost effectiveness (cost of hospitalization episode), and health-related quality of life.

Authors also plan to perform an internal pilot study to assess site and participant recruitment and consent rate, as well as compliance with protocol (procalcitonin assessments) and the ability to gather primary outcome data. Finally, authors will also perform a qualitative evaluation of the process of participating in the study.

My major questions/comments relate to clarification of the timing of randomization, patient eligibility criteria, and the intervention. Other minor comments are listed below.

Comment	Response
Participants:	
Is the diagnosis of the infectious processes based on the treating team opinion or will they be adjudicated by the research team? If adjudication will be used, more information on how it will be performed is needed.	Have added to Methods section (p.6) Suspected or confirmed SBI is diagnosed by the clinical team.
Please provide definitions for "suspected" and "confirmed" SBI.	This has been added (p.3) Defined previously ⁶ . Irwin AD, et al. Predicting Risk of Serious Bacterial Infections in Febrile Children in the Emergency Department. Pediatrics . 2017 Aug;140(2):e20162853. doi: 10.1542/peds.2016-2853.
The rationale for excluding patients with bacterial meningitis is not presented. However, patients with uncomplicated culture-negative meningitis, which may be caused by viruses, will	This has been added to Table 1. The NICE bacterial meningitis guideline has predefined recommendations for bacterial

be included. Bacterial meningitis is an important severe infection in the pediatric patient population for which guidance regarding duration of IV antibiotic/total duration of antibiotic treatment would be very useful.	meningitis, therefore the PCT test will not influence duration of Iv antibiotics. Meningitis (bacterial) and meningococcal septicaemia in under 16s: recognition, diagnosis and management, Clinical guideline [CG102]Published: 23 June 2010 Last updated: 01 February 201
The definition of complicated bone and joint infections should be provided	This has been added to Table 1. Complicated bone and joint infections are when the is chronic and/or related to a fracture or fixation device or prosthesis or implant. Chronic osteomyelitis presents six or more weeks after bone infection and is characterised by the presence of bone destruction and formation of sequestra.
Identification and screening (Page 8, line 30):	
More detail in this section must be provided. Please describe how potentially eligible participants at different study sites will be identified.	This has been clarified (p.8) Member of the research team visiting the wards where children with SBI are admitted to assess eligibility and screening admissions lists
Intervention (Page 6, line 41):	
What is the intended timing of randomization? Will patients be eligible to be randomized at any time of the antibiotic treatment (e.g., day 5) or during a specific time window?	This has been clarified (p.10) Children will only be randomised if the clinician expects IV antibiotics will be prescribed for longer than 48 hours. This will typically be between 20-48 hours after admission, to fit with clinical work flow of ward rounds and phlebotomy times for routine blood tests.
Randomization: more details are needed. Based on the protocol, it seems that central randomization will be stratified by age group and site. What are the age groups that will be used? A random element will be added to the randomization algorithm: please clarify how this will be done.	The age groups and random element have been specified in a separate randomisation protocol, but this information should remain hidden from anyone involved in recruitment to minimise the risk of subversion. Hence it cannot be disclosed in this protocol paper. We have added a sentence to the paper explaining this (p.10).
Blinding: based on the protocol, I presume that patients/families and treating team will not be blinded to the intervention. However, will members of the research team be blinded for study group allocation (outcome assessors, statistician, etc)?	This has been added (p.13). The Trial Statistician will be blinded to the study group allocation.
Procalcitonin measurement: authors state that procalcitonin will be measured at randomization and every 1-3 days. Will the frequency of procalcitonin test be defined by the treating team or will it be based on the study protocol? More clarification is needed.	This has been clarified in the methods (p.7) to align with clinical workflow and routine laboratory testing where possible.

In addition, procalcitonin will be measured using the bioMérieux VIDAS platform. Do all sites have access to it? What is the test turnaround time? Will the same machine models and calibration parameters be used at all sites? What is the measuring range of the platform that will be used? Will you use serum or plasma?	This has been clarified in the Methods section (p.7). It is a prerequisite that participating sites have access this platform to take part in the trial. This is a semiautomated immunoassay system based on Enzyme Linked Fluorescent Assay principles. Calibration is performed in line with manufacturer's guidelines. It is simple and flexible to use and gives results in 20 minutes.
Procalcitonin algorithm: the rationale behind the procalcitonin algorithm that will be used in the study should be discussed in detail. It is important to specify that the thresholds that will be used were obtained from previous adult studies and not from pediatric patients.	This has been added to the Intervention section (p.7) Values are based on our now published work which is the largest study to prospectively assess the performance of multiple biomarkers of SBI in a heterogeneous cohort of critically ill children and uniquely profiles longitudinal biomarker changes within the cohort. Nielsen MJ, et al. Procalcitonin, C-reactive protein, neutrophil gelatinase-associated lipocalin, resistin and the APTT waveform for the early diagnosis of serious bacterial infection and prediction of outcome in critically ill children. PLoS One. 2021 Feb 5;16(2):e0246027. doi: 10.1371/journal.pone.0246027. PMID: 33544738; PMCID: PMC7864456.
Outcomes:	
Mortality: please define the type of mortality that will be studied (e.g., hospital mortality, 30-day mortality, etc).	Mortality is defined in the protocol as "death for any reason in the 28 days following randomisation" (see Table 2) (p.24).
Hospital-acquired infections (HAI): how will the diagnosis of such infections be made? Will they be based on the treating team diagnosis? Or will they be adjudicated by the local infection control team? Are the HAI definitions used in different participating sites the same (or at least similar)?	HAI is defined by the clinical team. This has been clarified in secondary outcomes measures (p.8) The definitions used are the same across all sites.
Suspected adverse drug reactions: it is very difficult to establish a causal relationship for an adverse reaction, especially when patients are taking multiple medications. How will this outcome be adjudicated?	This has been added (p.8). The Liverpool ADR Causality Assessment Tool (LCAT) arranges ten modified questions with their dichotomous responses into a flowchart to arrive at one of four outcome categories: 'definite', 'probable', 'possible', or 'unlikely' . It has high interrater agreement .

	Gallagher RM, Kirkham JJ, Mason JR, Bird KA, Williamson PR, Nunn AJ, et al. Development and inter-rater reliability of the Liverpool adverse drug reaction causality assessment tool. PLoS One. 2011;6(12):e28096. pmid:22194808
Sample size (Page 11, line 11):	
The authors have referred to the primary outcome outcome throughout the manuscript as “duration of IV antibiotics,” but could consider renaming it, e.g., “time in days until IV antibiotic cessation”, to more clearly align with the statistical analysis (Cox model) chosen.	This does not make a difference statistically so we have not changed the wording.
Non-inferiority margin: non-inferiority margin for a composite safety outcome that includes mortality was set at 5%. Earlier procalcitonin RCTs were criticized because the non-inferiority margin for mortality was considered too high. More clarification regarding the rationale behind the 5% non-inferiority margin is needed.	Previous the non-inferiority margin of 8-10% were used in adults. In this study, we chose a smaller non-inferiority bound of 5%, which accommodated the combined non-inferiority margin of three safety outcomes (unscheduled admissions/re-admissions, unscheduled readmission to PICU and mortality).
In line with the previous comment, have the authors considered also reporting measures on the time scale (e.g., comparing median durations)? This can be useful in cases when the event of interest (switching IV to oral) will eventually occur in all patients.	The reporting of median (interquartile range [IQR]) of duration outcomes were added into the statistical analysis section (p.14).
Statistical analysis:	
As the hazard ratio depends on length of follow-up, authors should clarify what follow-up duration (e.g., 28 days) is being considered for the Cox regression for the co-primary outcome (time to switch from IV to oral).	The study aims to follow-up all participants until their IV antibiotics are stopped, hence no follow-up duration is specified. We do not expect any right-censoring for this variable. NB, the primary outcome is not time to switch from IV to oral (which is a secondary outcome) but rather time to stop IV, which does not necessarily imply a switch to oral but can mean stopping antibiotics altogether.
Secondary outcomes: The secondary outcomes listed on Page 7 include 1) time to switch from broad to narrow spectrum antibiotics, and 2) hospital acquired infection up to day 28. However, the statistical analysis for these outcomes is missing.	These two outcomes have been added to the ‘Secondary outcomes’ subsection (p.14) and to Table 6 where analyses of secondary outcomes are described (p.27).
Interim analysis: it is not clear if an interim analysis will be performed. Information about when it will be performed, as well as the	This has been clarified (p.13). No interim analysis will be performed.

statistical methods that will be used for it, must be provided.	
Qualitative study – methods: what is the theoretical framework that will be used to analyze the qualitative data? How long after the intervention will interviews be conducted? How will participants be selected? How will final sample size be defined?	It is already stated that Thematic analysis will be used (p.16). Further info added (p.16). Interviews will be conducted after the participant reaches their day 28 follow up. A purposive sample strategy will be employed to include parents from both intervention and control arms and inclusion of different site. A purposive sample strategy will be developed to address representation from up to 5 different sites and variation in health professional role (e.g. ward nurse. consultant, research nurse etc.) With regards to the sample size for both health professionals and parents, the qualitative researcher(s) will make pragmatic decisions along with the research team regarding when enough is known about certain themes (I.e. data saturation has occurred).
Internal pilot: It is unusual to perform a pilot study embedded in the final RCT. It is commonly performed before the RCT to generate data on study feasibility. Questions arise regarding the internal pilot study: when are the planned dates for the internal pilot? How many and which sites are being included in the pilot trial? What happens if data show low proportion of internal pilot outcomes (e.g., low proportion of patients undergoing procalcitonin measurement, low proportion of patients for whom primary outcome data were collected, etc)? Will the RCT be considered as non-feasible? More clarification about how the results of the internal pilot study will be used in needed.	A section on the internal pilot has been added (p.8). An internal pilot phase will be conducted over the first eight months of the recruitment period with six lead sites. Predefined progression criteria will be used to assess feasibility to progress to the full trial, such as site and patient absolute recruitment and consent rate, proportion of patients undergoing PCT assessments and the ability to collect primary outcome data.
Minor comments	
Methods - Setting: how many hospitals will be participating in this RCT? It is not clear in the protocol.	This has been added (p.6) approximately n=10
Page 3, line 17: CRP not previously defined.	This has now been defined (p.3)
Page 6, line 52: Authors should state for ethical reasons the amount of blood sample needed from each child to achieve the 200µl of plasma.	This has been clarified (p.7).

	An additional 1ml (minimum 0.5ml) lithium heparin samples will be collected for PCT analysis.
Secondary outcomes (Page 7, line 43): Please provide a reference for both the Liverpool causality assessment tool and the CHU9D instrument, and a description of their reliability/validity.	These references have been added.
Page 8, line 14: Please describe how the paper copies will be kept confidential and for how long.	This has been clarified (p.9). Essential documents will be kept securely in a locked cupboard, and at the end of the trial, will be archived at an approved external storage facility for 10 years.
Page 9, line 6: Will authors seek to replace subjects who withdraw in order to maintain adequate sample size?	It is not intended to replace participants who withdraw as the recruitment target has already been inflated to account for 10% loss to follow-up.
Page 9, line 17: Please define age groups.	We have clarified that age groups will not be disclosed (p.10). To further minimise the risk of subversion, the specific age groups and the random element will not be disclosed to those involved in recruiting participants until the end of the recruitment period.
Page 12, line 21: Small typo. Please remove “. are observed”.	This has been corrected.
Page 23, line 27: Missing a word after “Antibiotic type, dose, duration, including...”	Word ‘including’ has been removed. Table 3 (p.25).

VERSION 2 – REVIEW

REVIEWER	Shehabi, Yahya Monash University
REVIEW RETURNED	05-Aug-2021
GENERAL COMMENTS	Thank you for asking to review this nicely written protocol. I have some minor comments:  1. In the abstract, there is a reference to recruitment breakup due to COVID. Please clarify if the study is already up and running at the time of submission. If so, please provide dates for commencement of recruitment, stopping and restart dates. 2. The co-primary outcome can be confusing; A single primary outcome of duration of IV antibiotics (Total days whilst in hospital or day 28) would have been preferred.

	a. The other safety outcomes could be used as such without inclusion into a composite outcome, in particular with low event rates. 3. This would have allowed for a simpler same size calculation based on superiority according to median days/hours of TOTAL IV antibiotics excluding interruptions. 4. It would be helpful to measure PCT in all patients at randomisation and keep the control group PCT blinded. This would help assessment of baseline data for any imbalances. 5. Sub-group analysis could include a. Different age groups (72 hrs to 18 years is a very wide range). b. Gram negative vs gram positive organisms. c. Vasopressor vs no vasopressor dependency. d. Acuity scoring subgroup. 6. Finally, the cut-off of 0.25 for stopping antibiotics might be too high. The normal PCT is 0.1 and there will be kids with chest or URTI who could have a PCT between 0.1 and 0.25. 7. A measure of compliance with PCT algorithm should be included.
--	---

VERSION 2 – AUTHOR RESPONSE

Comments from Reviewer 1:

1. In the abstract, there is a reference to recruitment breakup due to COVID. Please clarify if the study is already up and running at the time of submission. If so, please provide dates for commencement of recruitment, stopping and restart dates.

Response: This information has been added to the abstract and in main text 'Methods and Analysis' (p.2)

Recruitment commenced in June 2018 and paused briefly between March-May 2020 due to the COVID-19 pandemic.

2. The co-primary outcome can be confusing; A single primary outcome of duration of IV antibiotics (Total days whilst in hospital or day 28) would have been preferred.

a. The other safety outcomes could be used as such without inclusion into a composite outcome, in particular with low event rates.

Response: The study has already commenced so we cannot alter the design.

We direct the reviewer to the below paper on the use of co-primary outcomes for trials of antimicrobial stewardship interventions that is referenced in this protocol paper in 'Sample size' (p.12):

Gillespie D, Francis N, Carrol E, et al. Use of co-primary outcomes for trials of antimicrobial stewardship interventions. *The Lancet Infectious Diseases* 2018;18(6):595-97.

a. Each safety outcome that is part of the composite outcome will also be analysed separately as a secondary outcome.

3. This would have allowed for a simpler same size calculation based on superiority according to median days/hours of TOTAL IV antibiotics excluding interruptions.

Response: See comment above.

4. It would be helpful to measure PCT in all patients at randomisation and keep the control group PCT blinded. This would help assessment of baseline data for any imbalances.

Response: The study has already commenced so we cannot alter the design.

As described in 'Primary Outcome' (p.14) the Trial Statistician will be blinded to the study group allocation.

Due to the large sample size of 1942 in combination with the randomised design, no major baseline imbalances are expected, but this will be checked as part of a descriptive data analysis of other baseline variables.

5. Sub-group analysis could include:

- a. Different age groups (72 hrs to 18 years is a very wide range).
- b. Gram negative vs gram positive organisms.
- c. Vasopressor vs no vasopressor dependency.
- d. Acuity scoring subgroup.

Response: We have amended a sentence in 'Sub Group Analysis' (p.12) to clarify that several sub-group analyses will be performed.

Several exploratory sub group analyses, including one split by the organ system of the infection (i.e. lower urinary tract, lower respiratory, intra-abdominal, bacteraemia, skin and soft tissue, etc), will be pre-specified in the Statistical Analysis Plan.

6. Finally, the cut-off of 0.25 for stopping antibiotics might be too high. The normal PCT is 0.1 and there will be kids with chest or URTI who could have a PCT between 0.1 and 0.25.

Response: The thresholds have been defined by previous data as described in 'Intervention' (p.7).

We direct the reviewer to the below paper on PCT and other biomarkers for the early diagnosis of serious bacterial infection and prediction of outcome in critically ill children:

Nielsen M, Baines P, Jennings R, et al. Procalcitonin, C-reactive protein, neutrophil gelatinase-associated lipocalin, resistin and the APTT waveform for the early diagnosis of serious bacterial infection and prediction of outcome in critically ill children. PLoS One 2021;16(2)

In this longitudinal biomarker study, we showed that median PCT in the non-SBI group was up to 1.5, and most were in the 0.25-0.5 range. A cut off of 0.1 would have been too low, and would have included many without infection.

7. A measure of compliance with PCT algorithm should be included.

Response: We have amended a sentence in 'Primary Analysis' (p.12) to clarify that adherence will be measured.

Adherence to the PCT algorithm will be recorded, and per-protocol and complier average causal effect (CACE) analysis will be undertaken to test the treatment effects to patients with fully utilised PCT algorithm (Table 5).